# Disclosing political partisanship polarizes first impressions of faces

**Brittany S. Cassidy** [1]◉*, **Colleen Hughes** [2]◉, **Anne C. Krendl**[3]

**1** University of North Carolina at Greensboro, Greensboro, NC, United States of America, **2** Department of Neurology and Neurosurgery, Montreal Neurological Institute, McGill University, Montreal, Canada, **3** Indiana University, Bloomington, Indiana, United States of America

◉ These authors contributed equally to this work.
* bscassid@uncg.edu

**Data Availability Statement:** Additional methods, data, and code for all experiments are available at the Open Science Framework: https://osf.io/9khta.

**Funding:** This research was supported by grant numbers KL2TR002530 and UL1TR002529 (A.

## Abstract

Americans' increasing levels of ideological polarization contribute to pervasive intergroup tensions based on political partisanship. Cues to partisanship may affect even the most basic aspects of perception. First impressions of faces constitute a widely-studied basic aspect of person perception relating to intergroup tensions. To understand the relation between face impressions and political polarization, two experiments were designed to test whether disclosing political partisanship affected face impressions based on perceivers' political ideology. Disclosed partisanship more strongly affected people's face impressions than actual, undisclosed, categories (Experiment 1). In a replication and extension, disclosed shared and opposing partisanship also engendered, respectively, positive and negative changes in face impressions (Experiment 2). Partisan disclosure effects on face impressions were paralleled by the extent of people's partisan threat perceptions (Experiments 1 and 2). These findings suggest that partisan biases appear in basic aspects of person perception and may emerge concomitant with perceived partisan threat.

## Introduction

Political polarization in the United States has been a central focus of social science research for several decades [e.g., 1–3]. Americans have growing alignment on within-political party opinions ranging from gun control to same-sex marriage [4,5] to the extent that political ideology predicts policy preferences almost three times better than demographic factors like education [6]. Inherent to such polarization is intergroup tension. Indeed, conservatives and liberals are similarly intolerant toward each other [7], make negative attributions about groups whose values are inconsistent with their own [8], and avoid people who do not share their values [9,10]. Because critical societal challenges require bipartisan cooperation to address them [e.g., COVID-19; 11], identifying ways in which such polarization emerges has considerable utility. Although there are many ways that political ideologies may affect interpersonal behavior, the current investigation focused on face impressions, a well-studied aspect of person perception [12,13]. Face impressions affect how people behave toward others [e.g., 14] and become polarized based on incoming information [e.g., 15]. Identifying how simple partisan cues affect face

Shekhar, PI) from the National Institutes of Health, National Center for Advancing Translational Sciences (https://ncats.nih.gov/), Clinical and Translational Sciences Award to A.C.K. The authors declare no conflicts of interest. The funders had no role in study design, data collection and analysis, decision to publish, or preparation of the manuscript.

**Competing interests:** The authors have declared that no competing interests exist.

impressions may thus be useful to better characterize rising political sectarianism in America [16].

Recent work suggests that people devalue facial cues from targets who are ideologically dissimilar from them. Such work has been interpreted through the lens of attitudinal dissimilarity, whereby people respond negatively to dissimilar others [e.g., 7]. In one study [17], people viewed a dating profile featuring a face and limited information about the target (e.g., personality traits). Later, the target's partisanship was disclosed. Perceiver conservatism related to liking a conservative target more and liking a liberal target less after disclosure. This pattern aligned with work showing that people find target individuals as less physically attractive when they hold dissimilar political candidate preferences [18], highlighting a role of partisan dissimilarity in changing face impressions. Although these studies suggest that partisanship impacts face impressions, one limitation is that they focus on ideology. That is, they do not examine other factors affecting face impressions in parallel that could guide manipulations in future work to establish causal mechanisms for polarized partisan impressions. Further, that these studies were conducted in the context of forming romantic relationships makes it unclear whether the resulting polarization reflects a general effect or one limited to a specific motivational context. These questions are important given that face impressions affect the extent to which people cooperate with others [e.g., 19]. To this end, we present two experiments testing whether disclosing political partisanship polarizes face impressions in the absence of other information. Further, we explore whether partisan threat parallels expected ideology effects on this polarization.

Prior work supports that disclosed partisanship may polarize face impressions across contexts. For example, people more negatively perceive faces paired with negative versus neutral group labels [20] and treat outgroup faces in a negatively prejudicial way [21]. These findings extend work showing that visible group-associated cues elicit negative bias [e.g., on the basis of race; 22] by suggesting that labels simply implying that target individuals differ in group membership and values from perceivers polarize face impressions. Notably, ideological partisanship is a salient marker of relative value dissimilarity when disclosed to perceivers [e.g., 23,24]. Illustrating negative effects of this dissimilarity on social cognition, political partisanship elicits biases along party lines similar to racial biases [24–26], often outweighing other group memberships to predict bias [27].

Although some work suggests that partisanship is a relatively concealable aspect of identity [see 28], other work shows that people are relatively accurate at identifying political affiliation in the absence of explicit information [29,30]. At the same time, explicit partisan labels (e.g., Democrat) polarize impressions in contexts where romantic interests are salient [17] and can be randomly assigned to targets to elicit bias [31]. This work raises the possibility that explicitly disclosed partisanship may polarize impressions to a greater extent than cues that people may naturally detect. This possibility is important to study because people can be mischaracterized as belonging to a negatively evaluated group and incur negative bias [e.g., 32]. We hypothesized that pairing faces with partisan labels, irrespective of the implied veracity of those designations, would affect perceivers' impressions more than actual target partisanship not explicitly disclosed to perceivers (Experiment 1).

If disclosed partisan labels strongly affect person perception based on perceivers' ideological partisanship, they should also *change* first impressions after their disclosure. Thus, also of interest was whether simply pairing partisan labels with faces, irrespective of the accuracy of those pairings, modulated face impressions. To this end, Experiment 2 was a replication and extension of Experiment 1 in which people evaluated faces before and after target partisanship disclosure to measure impression updating. Prior work has not systematically examined such changes. Although some work suggests that face impressions are resilient to disclosed new

information [e.g., 33], other work indicates that people update face impressions depending on *what* information is disclosed. For example, although facial untrustworthiness elicits negative impressions, disclosing salient positive behaviors results in more positive impressions [15]. This finding aligns with work showing that impressions are updated after salient behaviors are disclosed [34], as well as work showing that impressions based on implicit cues are malleable based on explicit and diagnostic incoming information [35].

One possibility is that arbitrary partisan labels may polarize impressions once they are disclosed, irrespective of their accuracy and based on perceiver partisanship. That is, labeling someone as having the opposite partisanship as the perceiver should elicit negative impression change (i.e., impressions becoming more negative), whereas perceived shared partisanship should elicit positive impression change (i.e., impressions becoming more positive). Such patterns would extend work showing favoritism and, sometimes, derogation, based on group membership [36–39] from a romantic [17] to a more general context and show that simple partisan labels in the absence of other partisan information can powerfully affect impressions.

Showing that disclosed partisanship strongly affects face impressions, Experiment 1 tested whether accurately *and* inaccurately disclosed partisanship affects impressions more strongly than accurate, yet undisclosed, partisanship. Replicating and extending this finding, Experiment 2 tested the prediction that disclosing partisanship *changes* face impressions. Because opposing partisans are negatively evaluated [16], we expected the direction of impression change to be based on perceivers' ideological partisanship. Recognizing that partisans at both ends of the ideological spectrum express negative bias against ideologically dissimilar people [7], we expected similarly polarized biases from people identifying as more conservative and more liberal.

Although our main goal was to identify partisanship-based effects of disclosed partisanship on face impressions, an open question regarded what factors produce parallel effects. We examined this question on an exploratory basis. Recent work suggests that negative trait attributions of opposing partisans relate to threat opposing partisans are perceived to pose [40], a pattern consistent with intergroup threat theory [41]. Because the extent to which politically salient stimuli affect attitudes depends on their eliciting threatening feelings [42], it seemed plausible that patterns of perceived partisan threat on face impressions of disclosed partisans would parallel effects of perceiver political ideology. Supporting a connection between perceived threat and trait impressions of faces, recent work using visual cues showed that threatening contexts affect facial trustworthiness impressions more than other contexts [43]. If the presence of opposing partisan labels is threatening, the extent of partisan threat perceived from one party relative to another should affect face impressions and how they change like perceiver ideology is expected to affect them.

To further assess partisan disclosure effects on face impressions, we thus explored whether the expected patterns shown for perceivers' face impressions were paralleled by the extent of their self-reported perceived partisan threat. That is, if perceivers evaluated opposing partisans as being especially threatening relative to ingroup partisans, then they should especially favor similar over opposing partisans (Experiment 1) and accordingly change the valence of their face impressions based on disclosed target partisanship (Experiment 2).

## Experiment 1

Although people can often detect political partisanship from faces [29], disclosed group labels can override naturally occurring ones to elicit biases [44]. Disclosing partisan labels, irrespective of their accuracy, may thus affect impressions more than actual partisanship that, albeit potentially detectable, is not disclosed. We expected the direction of these effects to emerge

based on perceivers' political partisanship. Experiment 1 used impressions of unfamiliar political candidate faces to test these possibilities.

Impressions in this task involved asking people to select the more likable and competent of two unfamiliar faces (one Republican and one Democrat). These traits were selected because they are core dimensions of person perception [45] reflecting separable ways in which people stereotype others [46]. Devaluing these traits would complement distinct ways of deriding opposing partisans that are becoming more commonplace in the United States [16]. Partisan disclosure was manipulated between-subjects. In one task version, partisanship was not disclosed. Here, actual partisanship was expected to polarize impressions. In another version, partisan labels were paired with faces. These labels were accurate (i.e., consistent with actual partisanship) or inaccurate (i.e., reflecting an opposing partisanship). We expected perceiver political ideology (reflecting their own partisanship) to exacerbate intergroup bias (i.e., more frequently selecting faces labeled with shared partisanship to be more likable and competent than non-labeled faces with shared partisanship). Such patterns would show that explicit (versus more implicit) partisan cues polarize face impressions based on perceiver partisanship.

## Method

**Participants.** The Indiana University Institutional Review Board approved all experiments. All participants provided written informed consent. Power analyses using the R-package *WebPower* [47] targeted 74 participants to detect a moderate perceiver political ideology effect (i.e., a 20% lower probability of more conservative participants choosing a Democrat as the more positive of a pair of faces) with 80% power and $\alpha$ = .05. Because disclosure was manipulated between-subjects, we doubled the target sample to ensure enough participants in each version. We oversampled to account for exclusions and to increase the likelihood of a wide range of political ideologies. Of 185 undergraduates recruited from a large Midwestern university in the United States, we excluded four. Two did not complete the partisanship characterization measures (see below) and two failed the manipulation check (see below). The analyzed sample comprised 181 undergraduates ($M_{age}$ = 18.53 years, $SD$ = .81; 128 female; 143 White, 22 Asian, 8 Black, 3 multiple, 2 unknown; 10 Hispanic). See https://osf.io/9khta/?view_only=65e52204b50d492b975001825d2f4efc for additional methods and results (e.g., Supplementary Information.docx), data, and code for all experiments.

**Task.** One hundred ten pairs of neutrally expressive White male faces were drawn from databases of opponents in United States political races that have been used in past work [e.g., 48]. Each pair depicted one actual Republican and one actual Democrat who were opponents in a past political race. Thus, the pairs were pre-determined and the same across participants. Across pairs, half of the Republicans and Democrats had won. We counterbalanced whether the actual winner appeared on the right or left of the screen within task versions. Like prior work [48], we did not tell participants the pictures were of politicians.

In both experiments, the task was presented using E-Prime 2.0. Self-paced competence and likability evaluations were made over two evaluation-specific blocks of 110 trials each presented in a counterbalanced order. Pair presentation order was randomized. Before each block, participants saw the evaluation they would make ("You will now choose which of two faces is the more competent [likable]"). Each trial comprised a prompt ("Which person is the more competent [likable]") above two side-by-side faces. Participants selected whether which face appeared more competent [likable]. There was a 250ms blank screen between trials. To make partisanship the most salient difference within each pair, we did not include female faces.

Three task versions counterbalanced disclosed political partisanship on a between-subjects basis. In one version, partisanship was not disclosed. Because actual partisanship was known,

we could determine when actual Republicans or Democrats were selected. The second two versions disclosed partisanship for each face via a red border indicating a Republican and a blue border indicating a Democrat. Of these two versions, one had the left and right faces labeled, respectively, as Republican and Democrat. In the other, left and right faces were labeled as, respectively, Democrat and Republican. Participants did not explicitly categorize partisanship. Rather, we measured the frequency that a disclosed Republican or Democrat was evaluated as the more competent [likable] of the pair. Accuracy (correct or incorrect) of the partisanship disclosed for each face was thus counterbalanced across these versions, allowing us to test whether disclosure affected impressions irrespective of veracity.

Immediately after the task, participants disclosed if they recognized any faces. If they said yes (N = 66), disclosed who they thought they recognized. Our a priori exclusion criterion was to exclude any participants who accurately identified faces. No participants, however, did so. At the end of task versions with disclosed partisanship, analyzed participants accurately verified representative colors, which served as a manipulation check.

**Partisanship characterization.**  We collected partisanship characterization measures in a random order after the task.

*Perceiver political ideology.* Participants indicated political ideology over four items (overall, social issues, economic issues, and foreign policy issues) on a scale ranging from 1 *[extremely liberal]* to 9 *[extremely conservative]*, similar to past work [e.g., 49]. Responses (Cronbach's α = .90) were averaged to create a composite political ideology score ($M = 4.80$, $SD = 1.91$). As a single continuous variable relating to partisan prejudice [27], we measured effects of disclosure on impressions of faces with regard to composite political ideology. Although relative ideology does not exactly match Republican and Democrat labels, these correlated concepts can determine partisanship effects on person perception [50]. Composite political ideology scores did not differ between task versions in which labels were and were not disclosed, $F(1,179) = .41$, $p = .52$.

*Perceived partisan threat.* Because perceived partisan threat contributes to political polarization [e.g., 40], we characterized the threat with which participants perceived partisans over four items: "How much of a threat do you think a person of the following party [Republican, Democrat, Independent/undecided] poses to you [society]?" and "How much of a threat do you think a person of the following party *who is also an elected official* poses to you [society]?" using scales ranging from 1 *[not at all]* to 7 *[very much]*. Responses toward each party (Cronbach's α at least .81) were averaged to create three composite threat scores. Within individuals, we also calculated the difference in Democrat minus Republican threat composites. We standardized (i.e., z-scored) these differences across our sample for exploratory analyses of perceived partisan threat on the anticipated partisan disclosure effect.

## Results

**Analytic strategy.**  Across experiments, the base R function lm was used for linear regressions. Mixed effects models were fitted using lme4 [51]. Model *p*-values were calculated using lmerTest [52]. Confidence intervals were calculated via the base R function confint. The emmeans package [53] was used to calculate the estimated marginal means and simple effects tests reported alongside the regression results. *P*-values for post-hoc tests were adjusted using Tukey method. When *t*-tests were employed and group variances were unequal, we used the Welch-Satterthwaite approximation for degrees of freedom.

**Characterizing partisan disclosure effects on face impressions.**  We first tested whether disclosing partisanship more strongly affected impressions than non-disclosed partisanship. Likability and competency choices (Republican = 0, Democrat = 1) were logistically regressed

on Trait (competent = 0, likable = 1), Task Version (disclosed labels = 0, non-disclosed labels = 1), Perceiver Political Ideology (standardized around the composite political ideology scores for the sample to have a mean of 0 and a standard deviation of 1), and their interactions as fixed effects. Models with different random effects structures were compared to determine best fit [54]. A first model included random intercepts for participants and face. A second allowed a Trait effect to vary by participants. Because fit did not differ between the models, $\chi^2(2) = .20$, $p = .91$, we report from the simpler model.

A higher probability of selected faces disclosed as having shared partisanship would support a disclosure effect. Main effects of Task Version (reflecting more selected Democrats in the disclosed versus non-disclosed task version) and Perceiver Political Ideology (reflecting fewer selected Democrats with higher perceiver conservatism) were qualified by a Task Version × Perceiver Political Ideology interaction that partially supported this hypothesis (Table 1A; Fig 1). We allow main effects to be interpreted within the context of this higher-order interaction. Although not further qualified by Trait, post-hoc tests are reported by Trait for completeness. See Table 2A for estimated marginal means.

We defined more liberal and more conservative participants as having standardized composite political ideology scores that were, respectively, one standard deviation below and above the sample mean. Consistent with our hypothesis that partisan disclosure would polarize impressions, more liberal participants were more likely to select disclosed versus non-disclosed Democrats as more competent, $OR = 1.36$, $z = 5.44$, $p < .001$, 95% CI [1.18, 1.57], and likable, $OR = 1.36$, $z = 5.41$, $p < .001$, 95% CI [1.17, 1.57]. Inconsistent with this hypothesis, however, no differences emerged for more conservative participants (one standard deviation above the mean composite political ideology score) when choosing the more competent, $OR = 1.03$, $z = .50$, $p = .96$, 95% CI [.89, 1.19], or likable, $OR = 1.02$, $z = .34$, $p = .99$, 95% CI [.88, 1.18] face.

**Characterizing partisanship effects on face impressions by the veracity of disclosed labels.** Because people can often detect partisanship from faces alone [e.g., 29], our next analyses concerned determining whether the veracity of disclosed labels affected polarized face impressions based on perceiver political ideology. First, we examined whether face impressions were polarized by *non-disclosed* partisanship. Among participants for whom party labels were not disclosed, however, perceiver political ideology did not affect face selections, $OR = .98$, $p = .49$, 95% CI [.92, 1.04].

**Table 1. Mixed effects model predicting selected faces in Experiment 1.**

| | Probability of Choosing a Democrat (1) relative to a Republican (0) | | | | | |
|---|---|---|---|---|---|---|
| | a. Perceiver Political Ideology | | | b. Partisan Threat | | |
| *Predictors* | *Odds Ratio* | *95% CI* | *p* | *Odds Ratio* | *95% CI* | *p* |
| (Intercept) | 1.11 | 1.01 – 1.21 | **.027** | 1.12 | 1.02 – 1.23 | **.016** |
| Task Version [Non-disclosed] | 0.85 | 0.78 – 0.91 | **< .001** | 0.84 | 0.77 – 0.91 | **< .001** |
| Trait [Likable] | 1.00 | 0.94 – 1.07 | .927 | 1.01 | 0.95 – 1.07 | .871 |
| Ideology (a) / Threat (b) | 0.85 | 0.80 – 0.90 | **< .001** | 0.89 | 0.84 – 0.94 | **< .001** |
| Task Version [Non-disclosed] * Trait [Likable] | 1.01 | 0.93 – 1.09 | .898 | 1.00 | 0.92 – 1.09 | .969 |
| Task Version [Non-disclosed] * Ideology (a) / Threat (b) | 1.15 | 1.06 – 1.24 | **< .001** | 1.12 | 1.03 – 1.22 | **.006** |
| Trait [Likable] * Ideology (a) / Threat (b) | 0.98 | 0.92 – 1.04 | .437 | 0.94 | 0.89 – 1.00 | .059 |
| Task Version [Non-disclosed] * Trait [Likable] * Ideology (a) / Threat (b) | 1.00 | 0.93 – 1.09 | .929 | 1.06 | 0.97 – 1.15 | .187 |

The Task Version reference condition is Disclosed and the Trait reference condition is Competent. Reflecting the parallel nature of these analyses, columns A and B use perceiver political ideology and the difference in perceived partisan threat from Democrats relative to Republicans, respectively, as predictors.

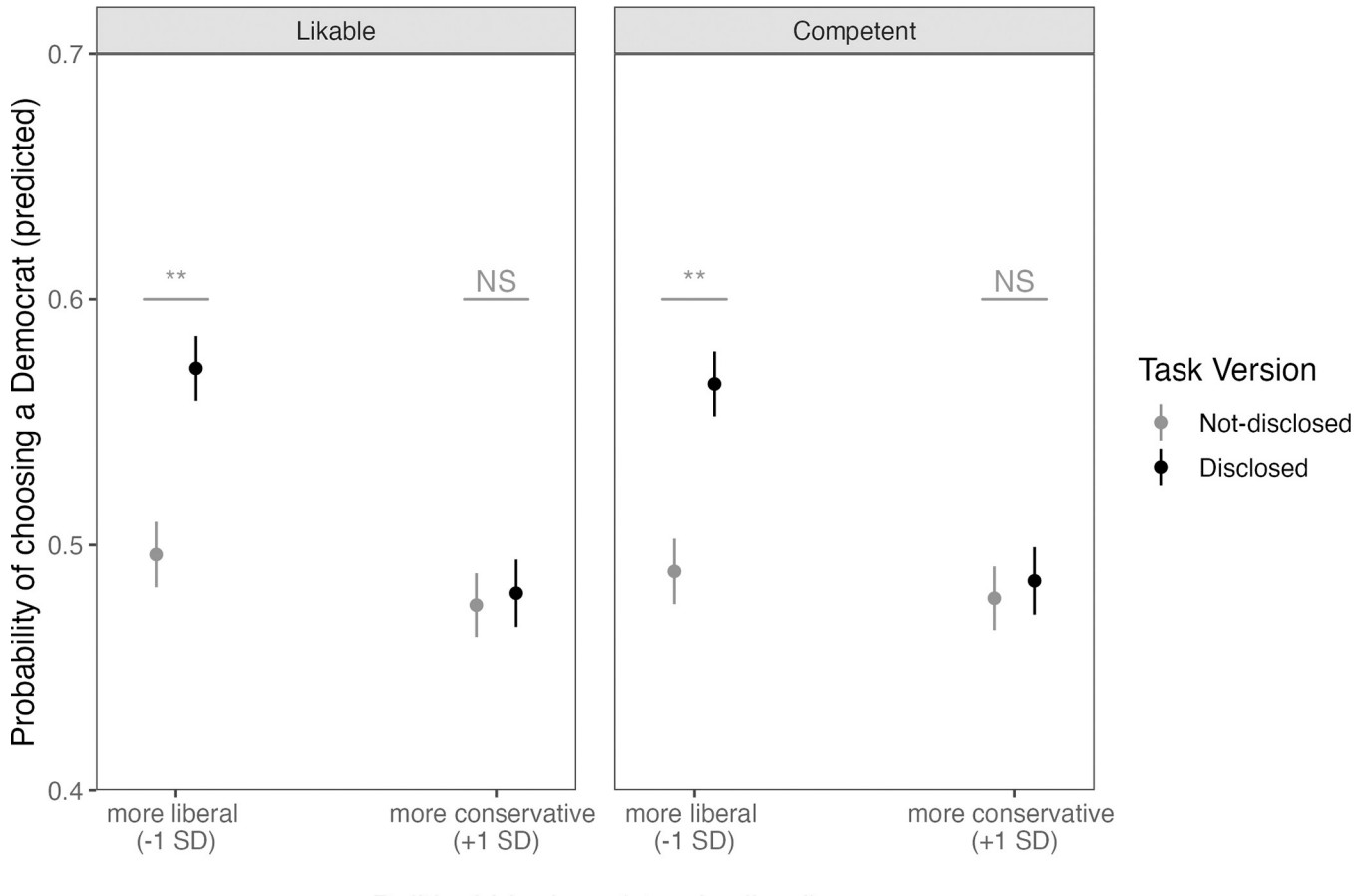

**Fig 1. Predicted probability of choosing a Democrat (vs. Republican) as a function of Trait (competent, likable), Task Version (not-disclosed or disclosed labels), and perceiver political ideology.** Points represent the condition means and whiskers represent the standard error of the mean. ** = < .001; NS = non-significant.

We next tested whether the veracity of *disclosed* partisanship affected face impressions. Likability and competency choices (Republican = 0, Democrat = 1) were logistically regressed on Trait (competent = 0, likable = 1), Disclosed Label Veracity (accurate = 1, inaccurate = 0), Perceiver Political Ideology, and their interactions as fixed effects (Table 3) among participants who saw the labels. The random effects structure included intercepts for participants and faces. A main effect of Perceiver Political Ideology reflected fewer selected Democrats with higher perceiver conservatism. A main effect of Disclosed Label Veracity reflected fewer selected Democrats with accurate versus inaccurate labels. This pattern may seem surprising because given the liberal skew of the sample, one might expect more accurately labeled Democrats because potential detections would match labels. Because differences between faces signal the likelihood of winning [48], it could also be that inaccurate labels resulted in more "Democrats" with positively interpreted facial cues. Suggesting disclosed labels affected impressions irrespective of their veracity, however, no interaction between Disclosed Label Veracity and Perceiver Political Ideology emerged.

**Exploring partisan disclosure effects on face impressions.** Exploratory analyses identified how the above-described partisan disclosure effects were paralleled by perceived partisan threat.

**Table 2. Estimated marginal means for Experiments 1–2.**

**a. Experiment 1**

| Trait | Task Version (labels) | Perceiver Political Ideology | Estimated Marginal Mean [95% CI] |
|---|---|---|---|
| Likable | Non-disclosed | Liberal | .50 [.47, .52] |
| Likable | Disclosed | Liberal | .57 [.55, .60] |
| Competent | Non-disclosed | Liberal | .49 [.46, .52] |
| Competent | Disclosed | Liberal | .57 [.54, .59] |
| Likable | Non-disclosed | Conservative | .48 [.45, .50] |
| Likable | Disclosed | Conservative | .48 [.45, .51] |
| Competent | Non-disclosed | Conservative | .48 [45, .50] |
| Competent | Disclosed | Conservative | .49 [.46, .51] |

**b. Experiment 2**

| Label | Time | Perceiver Political Ideology | Estimated Marginal Mean [95% CI] |
|---|---|---|---|
| undecided | Before Label | Liberal | 3.76 [3.56, 3.95] |
| undecided | After Label | Liberal | 3.87 [3.68, 4.07] |
| Democrat | Before Label | Liberal | 3.75 [3.54, 3.95] |
| Democrat | After Label | Liberal | 4.02 [3.81, 4.23] |
| Republican | Before Label | Liberal | 3.77 [3.56, 3.97] |
| Republican | After Label | Liberal | 3.13 [2.93, 3.34] |
| undecided | Before Label | Conservative | 3.64 [3.45, 3.84] |
| undecided | After Label | Conservative | 3.63 [3.44, 3.83] |
| Democrat | Before Label | Conservative | 3.69 [3.48, 3.90] |
| Democrat | After Label | Conservative | 3.31 [3.10, 3.52] |
| Republican | Before Label | Conservative | 3.63 [3.42, 3.83] |
| Republican | After Label | Conservative | 3.98 [3.77, 4.18] |

Liberal/conservative corresponds to -1/+1 SD above/below the mean on the composite political ideology score.

*Relations between perceiver political ideology and perceived partisan threat.* Preliminary correlations showed that perceiver political ideology differentially related to threat perceptions of Republicans, Democrats, and undecideds (Table 4). We next regressed perceiver political ideology on the perceived threat ratings of each party. The model was significant, $R^2 = .23$, $p <$

**Table 3. Mixed effects model predicting selected faces based on the veracity of partisan labels in Experiment 1.**

| Probability of Choosing a Democrat (1) relative to a Republican (0) | | | |
|---|---|---|---|
| Predictors | Odds Ratios | 95% CI | p |
| (Intercept) | 1.20 | 1.10 – 1.30 | < .001 |
| Trait [Likable] | 0.99 | 0.91 – 1.08 | .798 |
| Label Veracity [Accurate] | 0.85 | 0.78 – 0.92 | < .001 |
| Perceiver Political Ideology | 0.87 | 0.80 – 0.94 | .001 |
| Trait [Likable] * Label Veracity | 1.03 | 0.91 – 1.16 | .651 |
| Trait [Likable] * Perceiver Political Ideology | 0.98 | 0.90 – 1.06 | .568 |
| Label Veracity [Accurate] * Perceiver Political Ideology | 0.97 | 0.89 – 1.05 | .429 |
| Trait [Likable] * Label Veracity [Accurate] * Perceiver Political Ideology | 1.00 | 0.89 – 1.12 | .966 |

This analysis is based on the subset of participants for whom labels were disclosed (N = 82), and the Trait reference condition is Competent.

**Table 4. Means (M), standard deviations (SD), and intercorrelations (r) between political ideology, perceived threat, and political affiliation in Experiment 1 (lower diagonal) and 2 (upper diagonal).**

| Measure | Exp1 M [SD] | Exp2 M [SD] | 1 | 2 | 3 | 4 |
|---|---|---|---|---|---|---|
| 1. Political ideology | 4.80 [1.91] | 5.03 [1.79] | – | -.50** [-.64, -.33] | .48** [.31, .62] | -.00 [-.21, .20] |
| 2. Republican threat | 3.33 [1.74] | 3.24 [1.72] | -.48** [-.59, -.36] | – | .07 [-.14, .27] | .36** [.17, .53] |
| 3. Democrat threat | 2.58 [1.22] | 2.95 [1.34] | .24** [.09, .37] | .31** [.17, .43] | – | .30** [.11, .48] |
| 4. Independent/ undecided threat | 2.19 [1.11] | 2.42 [1.17] | .03 [-.12, .17] | .38** [.25, .50] | .56** [.45, .66] | – |

*$p < .05$

**$p < .01$. Numbers within brackets are the 95% confidence intervals. Higher values for political ideology indicate greater conservatism.

.001 (Table 5A). Patterns for more liberal participants were consistent with their face impressions. More liberal participants perceived Republicans ($M = 4.16$, $SE = .14$) as more threatening than Democrats ($M = 2.29$, $SE = .14$), $b = 1.87$, $t = 9.75$, $p < .001$, 95% CI [1.42, 2.32], and undecideds ($M = 2.16$, $SE = .14$), $b = 2.00$, $t = 10.43$, $p < .001$, 95% CI [1.55, 2.45]. They perceived Democrats and undecideds as similarly threatening, $b = .13$, $t = .68$, $p = .78$, 95% CI [-.32, .58]. Thus, more liberal participants showed both ideologically polarized face impressions and partisan threat perceptions. There were no 3-way interactions with Task Version, $b$s $< .42$, $p$s $> .12$.

Patterns for more conservative participants were also consistent with their face impressions. More conservative participants perceived Republicans ($M = 2.48$, $SE = .14$) versus Democrats ($M = 2.87$, $SE = .14$), $b = -.39$, $t = -2.02$, $p = .11$, 95% CI [-.84, .06] and Republicans versus undecideds ($M = 2.22$, $SE = .14$), $b = .26$, $t = -1.32$, $p = .39$, 95% CI [-.20, .71], as similarly threatening. However, they perceived Democrats as more threatening than undecideds, $b = .64$, $t = 3.34$, $p = .003$, 95% CI [.19, 1.10]. Thus, more conservative participants did not show ideologically polarized face impressions nor partisan threat perceptions.

*Relations between partisan disclosure effects and perceived partisan threat.* Next, we verified that the difference in partisan threat perceptions of Democrats and Republicans (see above) positively related to perceiver political ideology, $r(179) = .63$, $p < .001$. This relation validated that people self-reporting as more conservative found Democrats more threatening relative to Republicans. We therefore explored whether this difference in perceived partisan threat had similar partisan disclosure effects as perceiver political ideology on face impressions

**Table 5. Regression models predicting partisan threat perceptions (Republican, Democrat, undecided) from perceiver political ideology in Experiments 1 & 2.**

| Predictors | a. Experiment 1 | | | b. Experiment 2 | | |
|---|---|---|---|---|---|---|
| | Estimates | 95% CI | p | Estimates | 95% CI | p |
| (Intercept) | 2.19 | 2.00–2.38 | < .001 | 2.42 | 2.15 – 2.68 | < .001 |
| Political Party [Democrat] | 0.39 | 0.12–0.65 | **.005** | 0.54 | 0.16 – 0.91 | **.005** |
| Political Party [Republican] | 1.13 | 0.86–1.39 | < .001 | 0.83 | 0.45 – 1.20 | < .001 |
| Perceiver Political Ideology | 0.03 | -0.16–0.22 | .762 | -0.00 | -0.27 – 0.26 | .990 |
| Political Party [Democrat] * Perceiver Political Ideology | 0.26 | -0.01 – 0.52 | .060 | 0.64 | 0.27 – 1.02 | **.001** |
| Political Party [Republican] * Perceiver Political Ideology | -0.87 | -1.14 – -0.60 | < .001 | -0.86 | -1.23– -0.48 | < .001 |

The Political Party reference condition is undecided.

(Table 1B). Across traits, the difference in perceived partisan threat indeed qualified a Task Version effect on face impressions just as perceiver political ideology did. Participants who perceived Republicans as being more threatening than Democrats (i.e., participants one standard deviation below the mean threat difference score) were more likely to select disclosed versus non-disclosed Democrats as more competent, $OR = 1.34$, $z = 4.98$, $p < .001$, 95% CI [1.15, 1.56], and likable, $OR = 1.41$, $z = 5.88$, $p < .001$, 95% CI [1.22, 1.64]. Similar to the above-reported analyses with perceiver political ideology, no difference emerged for participants who perceived Democrats as being more threatening than Republicans (i.e., participants one standard deviation above the mean threat difference score) when choosing the more competent, $OR = 1.07$, $z = 1.11$, $p = .68$, 95% CI [.92, 1.24], or likable, $OR = 1.01$, $z = .15$, $p > .99$, 95% CI [.87, 1.17] face.

## Discussion

Disclosed partisanship polarized face impressions among more liberal, but not more conservative, perceivers. This pattern partially supported that disclosed partisanship polarizes impressions based on perceiver partisanship. These patterns emerged regardless of the disclosed label's veracity (e.g., it did not matter if an actual Republican was labeled as a Democrat). Thus, simply implying partisanship is enough to polarize face impressions. Although partisanship can be detected from facial cues alone [e.g., 29], non-disclosed partisanship did not polarize face impressions. Although it could be that participants did not detect partisanship from these faces, another possibility is that being asked to evaluate traits overrode undisclosed partisanship effects on face impressions in this task overall [see 48]. Future research may assess this possibility by addressing partisanship effects on face impressions when people are informed, for example, that they are evaluating politicians versus not.

These patterns emerged regardless of whether perceivers selected faces as more likable or competent. Partisanship thus polarizes impressions spanning well-studied primary dimensions of social perception capturing separable ways in which people stereotype others [46]. Future work may replicate these findings while focusing on reaction times to better understand why they emerged. For example, it could be that label disclosure enables people to require less evidence to say a similar relative to opposing partisans are competent and likable. However, it could also be that people more steeply accumulate evidence of competence and likability from similar relative to opposing partisans. Disentangling these findings using drift diffusion modeling [e.g., 55], can help clarify processes underlying face impressions polarized by partisanship.

Prior work suggests that perceived partisan threat drives ideological prejudice [40]. One possibility was thus that the lack of polarized impression from conservative perceivers would be paralleled by their not perceiving opposing partisans as threatening to the same extent as more liberal perceivers. Indeed, perceived partisan threat perceptions paralleled both more liberal and more conservative participants' face impressions. Here, more conservative participants did not perceive Democrats as more threatening than Republicans. More liberal participants, by contrast, perceived Republicans as significantly more threatening than Democrats. That more conservative participants perceived Democrats as more threatening than undecideds suggests their threat perceptions were not indiscriminately attenuated–a finding consistent with people treating undecideds more favorably than opposing partisans [26]. Moreover, the above-described findings replicated when replacing perceiver political ideology with perceived partisan. That perceiver political ideology and perceived partisan threat were strongly related is consistent with growing political sectarianism in the United States [16]. It also aligns with work showing that threatening contexts polarize valenced face impressions

[43]. Speculatively, simple partisan labels may be enough to provide the threat that polarizes first impressions.

Although threat ratings suggested that more conservative perceivers did not find Democrats as more threatening than Republicans, significant correlations emerged between perceiver ideology and partisan threat perceptions. What might have caused this inconsistency? One possibility may lie in the college-aged sample recruited for the experiment. College-aged students often show a bias to perceiving themselves as more conservative than they really are [56]. If the students identifying themselves as more conservative were indeed more liberal than they realized, it would allow for the possibility of threat perceptions less extreme than those of the students identifying as more liberal. Indeed, these biased perceptions of one's own partisanship are more pronounced for conservatives than for liberals [56].

Although these patterns provide some initial correlational evidence that perceived threat may drive impressions of similar to opposing partisans [e.g., 40], several possibilities remained open for exploration. First, it could be that more liberal and conservative perceivers differentially use partisan labels when forming face impressions. Indeed, Republicans endorse more "Republican-looking" candidates as being likeable and competent [57]. Second, more liberal and conservative perceivers could both be affected by partisan cues in their face impressions but start at different baselines when making them. If true, that would make partisan cue effects on face impressions difficult to detect in a task where impressions were measured using a binary choice. To consider these explanations, Experiment 2 was designed to replicate and extend Experiment 1.

## Experiment 2

Experiment 2 replicated and extended Experiment 1 by characterizing whether partisan disclosure elicited changes in face impressions. Here, we tested whether disclosing opposing partisanship negatively changed impressions and if disclosing shared partisanship positively changed them based on perceiver political ideology. By using a scale to characterize face impressions, we could measure whether more liberal and more conservative perceivers broadly differed in how they approached making face impressions and if their partisan threat perceptions paralleled their face impressions. Here, people saw a face and evaluated likability before and after the face was paired with a partisan label. Evaluation change from before to after disclosure quantified impression change. If more liberal and conservative perceivers similarly approach making face impressions, we expected people to have positive and negative impression change, respectively, toward faces disclosed as having shared and opposing partisanship. If partisan threat perceptions underscore face impressions when given explicit partisan cues, we expected impression change to mirror perceived partisan threat. That is, if more conservative individuals perceived Democrats as more threatening than Republicans (unlike Experiment 1), we expected that they would negatively change their impressions of disclosed Democrats. Such a pattern would suggest perceived partisan threat as concomitant process alongside partisan impression change.

In addition to Republican and Democrat targets, a subset of targets was identified as undecideds. This addition allowed us to make a more nuanced interpretation of impression change based on partisan disclosure and a potential parallel pattern in perceived partisan threat. Politically undecided people vary in their identification with Republicans and Democrats [58], making their partisanship ambiguous and providing a natural control. People over-exclude others from their ingroups [59,60], suggesting positive change might be reserved for ideologically similar targets. If over-exclusion elicits similar change toward any targets who do not share perceiver ideology, similarly negative impression change should emerge for disclosed opposing

partisan and undecided faces. This pattern would support change largely explained by shared attributes with perceivers. People, however, behave more favorably toward independent versus opposing partisans [26]. Another possibility is thus that stronger negative change will emerge for opposing partisan versus undecided faces among more extreme partisans because the former reflects a group especially derogated by partisans [16]. The latter relative to the former possibility would be more consistent with an expectation of partisan disclosure effects on face impressions paralleled by perceived partisan threat effects.

## Method

**Participants.**   Power analyses [47] targeted 77 participants to detect a moderate political ideology effect ($f^2$ = .15) on impressions before versus after disclosure with 80% power and α = .05. We oversampled for the same reasons as in Experiment 1. Of 101 undergraduates recruited from a large Midwestern university, we excluded seven for failing the manipulation check. The analyzed sample comprised 94 undergraduates ($M_{age}$ = 18.90 years, $SD$ = 2.43, 64 female, 77 White, 11 Asian, 3 Black, 1 multiple, 1 unknown; 3 Hispanic).

**Task.**   One hundred twenty neutrally expressive younger adult White faces (60 male and 60 female) were drawn from the PAL database [61] on the basis of attractiveness and trustworthiness norms [see 62]. Forty faces each were randomly selected for one of three group categories (i.e., Republican, Democrat, or undecided). Three task versions counterbalanced the partisan label (Republican, Democrat, or undecided) paired with each face on a within-subjects basis. Depicted faces' actual partisanship was unknown. Male and female faces were equally represented across the three categories. We included female faces because, unlike Experiment 1, people did not choose between two partisan faces. Two ANOVAs showed that male and female faces paired with each category did not differ on attractiveness or trustworthiness (all $F$s < 3.10, all $p$s > 0.08).

In each trial, participants first saw a face for 1000ms followed by a scale (1 *[extremely dislike]* to 7 *[extremely like]*). They were told their self-paced likeability evaluations should be based on the picture. Immediately following, they were told they would be provided information indicating the political partisanship of the individuals and that they would evaluate them again. In a second 1000ms face presentation, a colored border surrounding the photo denoted political partisanship. Republicans were denoted with red borders, Democrats with blue borders, and undecideds with yellow borders. Participants then made another self-paced likeability evaluation. There was a 500ms blank screen between each trial. Participants verified colors designations before and after the task, which served as a manipulation check.

**Partisanship characterization.**   Participants completed the same measures as in Experiment 1. We summarize data assessing perceiver political ideology and perceived threat here. The political ideology items (Cronbach's α = .88) were averaged to create a composite political ideology score ($M$ = 5.03, $SD$ = 1.79). The threat items for each party (Cronbach's α at least 0.85) were averaged to create three composite threat scores.

## Results

**Characterizing impression modulation by partisan disclosure.**   Likability evaluations were regressed on Time (before label = 0, after label = 1), Partisan Label (Dummy coded using "undecided" as the reference of 0: Republican, Democrat, undecided), Perceiver Political Ideology (standardized as in Experiment 1), and their interactions as fixed effects. Like Experiment 1, the reported model included random intercepts for participants and face and allowed a Partisan Label effect to vary by participants. This model fit better than one with only random

intercepts for participants and face, $\chi^2(5) = 645.53$ $p < .001$. A third model allowing a Partisan Label by Time interaction to vary by participants failed to converge.

As hypothesized, interactions supported positive and negative impression change based on Partisan Disclosure and Political Ideology (Table 6A; Fig 2). More liberal participants liked people less after seeing Republican borders, $b = -.63$, $z = -15.18$, $p < .001$, 95% CI [-.75, -.52], and more after seeing Democrat borders, $b = .27$, $z = 6.51$, $p < .001$, 95% CI [.15, .39]. Impressions did not change after seeing undecided borders, $b = .12$, $z = 2.78$, $p = .06$, 95% CI [-.00, .24]. More conservative participants liked people more after seeing Republican borders, $b = .35$, $z = 8.37$, $p < .001$, 95% CI [.23, .47], and less after seeing Democrat borders, $b = -.38$, $z = -9.01$, $p < .001$, 95% CI [-.50, -.26]. Impressions did not change after seeing undecided borders, $b = -.01$, $z = -.26$, $p > .99$, 95% CI [-.13, .11]. See Table 2B for estimated marginal means.

**Exploring partisan disclosure effects on face impressions.** *Relations between perceiver political ideology and perceived partisan threat.* Correlations again showed that perceiver political ideology related to different threat perceptions of Republicans, Democrats, and undecideds (Table 4). We then regressed standardized political ideology scores on the perceived threat ratings of each party. The model was significant, $R^2 = .23$, $p < .001$ (Table 5B). More liberal participants perceived Republicans ($M = 4.10$, $SE = .19$) as more threatening than Democrats ($M = 2.31$, $SE = .19$), $b = 1.79$, $z = 6.68$, $p < .001$, 95% CI [1.16, 2.42]; and undecideds ($M = 2.42$, $SE = .19$), $b = 1.69$, $z = 6.30$, $p < .001$, 95% CI [1.06, 2.32], but perceived Democrats and undecideds as similarly threatening, $b = -.10$, $z = .39$, $p = .92$, 95% CI [-.74, .53]. Thus, more liberal participants showed both ideologically polarized face impressions and partisan threat perceptions.

Critically, more conservative participants perceived Democrats as more threatening than Republicans, $b = 1.21$, $z = 4.48$, $p < .001$, 95% CI [.57, 1.85]. More conservative participants perceived Democrats ($M = 3.59$, $SE = .19$) as more threatening than undecideds ($M = 2.41$, $SE = .19$), $b = 1.18$, $z = 4.37$, $p < .001$, 95% CI [.54, 1.82], and perceived Republicans ($M = 2.38$, $SE = .19$) and undecideds as similarly threatening, $b = .03$, $z = .11$, $p = .99$, 95% CI [-.67, .61]. Thus, more conservative participants showed both ideologically polarized face impressions and partisan threat perceptions.

**Table 6. Linear mixed effects model predicting evaluations in Experiment 2.**

| Predictors | a. Perceiver Political Ideology | | | b. Partisan Threat | | |
|---|---|---|---|---|---|---|
| | Estimates | 95% CI | p | Estimates | 95% CI | p |
| (Intercept) | 3.70 | 3.56 – 3.84 | < .001 | 3.71 | 3.57 – 3.85 | < .001 |
| Label [Republican] | -0.00 | -0.11 – 0.10 | .971 | -0.01 | -0.12 – 0.10 | .857 |
| Label [Democrat] | 0.02 | -0.08 – 0.11 | .736 | 0.01 | -0.09 – 0.10 | .912 |
| Time [After Label] | 0.05 | -0.01 – 0.11 | .074 | 0.05 | -0.01 – 0.11 | .079 |
| Ideology (a) / Threat (b) | -0.06 | -0.20 – 0.08 | .405 | -0.00 | -0.14 – 0.14 | .972 |
| Label [Republican] * Time [After Label] | -0.19 | -0.28 – -0.11 | < .001 | -0.21 | -0.29 – -0.13 | < .001 |
| Label [Democrat] * Time [After Label] | -0.10 | -0.19 – -0.02 | **.012** | -0.09 | -0.17 – -0.01 | **.026** |
| Label [Republican] * Ideology (a) / Threat (b) | -0.01 | -0.12 – 0.09 | .829 | -0.02 | -0.13 – 0.08 | .665 |
| Label [Democrat] * Political Ideology | 0.03 | -0.06 – 0.12 | .536 | 0.02 | -0.07 – 0.12 | .657 |
| Time [After Label] * Ideology (a) / Threat (b) | -0.06 | -0.12 – -0.01 | **.032** | -0.01 | -0.07 – 0.05 | .656 |
| Label [Republican] * Time [After Label] * Ideology (a) / Threat (b) | 0.56 | 0.47 – 0.64 | < .001 | 0.53 | 0.44 – 0.61 | < .001 |
| Label [Democrat] * Time [After Label] * Ideology (a) / Threat (b) | -0.26 | -0.34 – -0.18 | < .001 | -0.34 | -0.42 – -0.25 | < .001 |

The Label reference condition is undecided and the Time reference condition is Before Label. Reflecting the parallel nature of these analyses, columns A and B use perceiver political ideology and partisan threat, respectively, as predictors.

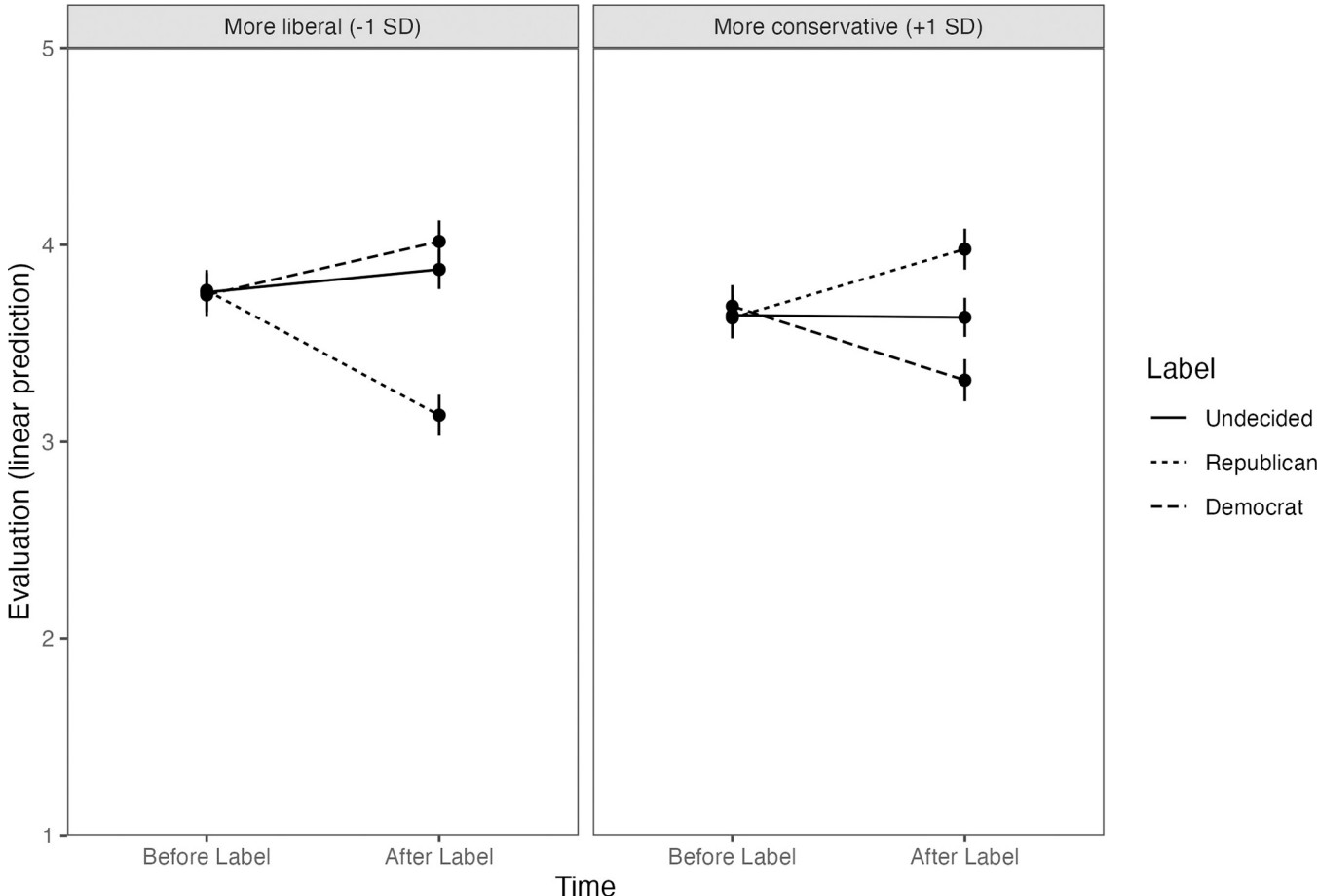

**Fig 2. Predicted likability as a function of Time, Label (party of evaluated face), and composite political ideology (+1 SD: More conservative; -1 SD = more liberal).** Points represent the condition means and whiskers represent the standard error of the mean.

*Relations between partisan disclosure effects and perceived partisan threat.* As in Experiment 1, perceiver political ideology positively related to the standardized difference in partisan threat perceptions of Democrats relative to Republicans, $r(92) = .72$, $p < .001$. Interactions supported positive and negative impression change based on Partisan Disclosure and Threat (Table 6B), again paralleling the results using perceiver political ideology. Participants who perceived Republicans as more threatening liked people less after seeing Republican borders, $b = -.67$, $z = -15.95$, $p < .001$, 95% CI [-.79, -.55], and more after seeing Democrat borders, $b = .31$, $z = 7.36$, $p < .001$, 95% CI [.19, .43]. Impressions did not change after seeing undecided borders, $b = .07$, $z = 1.55$, $p = .63$, 95% CI [-.05, .18]. Participants who perceived Democrats as more threatening liked people more after seeing Republican borders, $b = .36$, $z = 8.47$, $p < .001$, 95% CI [.24, .47], and less after seeing Democrat borders, $b = -.39$, $z = 9.32$, $p < .001$, 95% CI [-.51, -.27]. Impressions did not change after seeing undecided borders, $b = .04$, $z = 0.94$, $p = .94$, 95% CI [-.08, .16].

## Discussion

People positively changed their impressions of disclosed ingroup partisans and negatively changed impressions of disclosed opposing partisans based on their own political ideology. Just as salient behaviors change impressions of faces [e.g., 15], simply labeling faces with

partisan cues does too and that the extent of resulting polarization varies by people's partisan similarity to those cues. One question was whether negative change toward opposing partisans emerged because not sharing partisanship denotes a negative group or because opposing partisans specifically elicit negativity. Indeed, favoritism toward people sharing values can emerge without derogation toward those who do not [36, but see 63]. Because people behave more favorably to independents than to opposing partisans [26], examining impression change toward undecided and opposing partisan faces addressed these possibilities.

More conservative perceivers did not change impressions of undecideds after disclosure. Yet, they evaluated disclosed Republicans versus undecideds as more likable, $b = .35$, $z = 4.56$, $p < .001$, 95% CI [.13, .56]. These findings suggest favoring people with likely shared values in the absence of derogating undecideds, supporting that "ingroup love" motivates behavior over "outgroup hate" [e.g., 64]. Supporting this possibility, more liberal perceivers also did not change impressions of disclosed undecideds. These perceivers, however, did not differ in their impressions of disclosed Democrats versus undecideds. Speculatively, more conservative and liberal people may have different perceptions of the similarity of their group and undecideds, perhaps based on how they view the current polarized political climate. Future work may use a control condition with no denoted partisanship to examine this possibility.

Complementing Experiment 1, partisan threat perceptions paralleled impression change. Moreover, replacing perceiver political ideology with partisan threat perceptions in our model yielded the same patterns of impression modulation. Notably, whereas opposing partisans were perceived as more threatening than similar partisans, undecideds fell at a middle ground. These threat perceptions suggested that the more conservative participants in Experiment 2 may have been more likely to outwardly derogate Democrats [65], potentially explaining why conservative ideology affected impressions in Experiment 2, but not in Experiment 1. Correlations supported this explanation, as a positive relation between a more conservative ideology and perceptions of Democrats as threatening was double the size in Experiment 2 than in Experiment 1.

Further, that more conservative perceivers changed their impressions based on disclosed partisanship did not support the explanations that they simply used facial stereotypes [e.g., 57] more than explicit partisan cues when evaluating faces or that conservatives and liberals start their impressions in different places (e.g., starting more positively). Rather, these findings raise the possibility that partisan threat perceptions elicit changes to face impressions. Complementary recent work [40] suggests that perceived partisan threat may be more likely to mirror face impressions of partisans. The extent to which people dehumanize opposing partisans may thus depend, in part, on their perception of group-based threats from them [66].

## General discussion

The current work identified political ideology as affecting face impressions of disclosed partisans. Experiment 1 showed that disclosed partisanship more strongly affects face impressions than non-disclosed partisanship even when that disclosure is inaccurate. Experiment 2 showed that people change their impressions of disclosed partisans based on their own ideological partisanship. Across experiments, partisan disclosure effects on face impressions were paralleled by the extent of perceived partisan threat. These findings extend work showing partisan differences in face impressions in romantic contexts [17] to the general face impressions eliciting everyday approach and avoidance decisions [67]. They also build on work showing that perceived threat drives negative impressions of opposing partisans [40] by showing that partisan threat perceptions parallel partisan disclosure effects on face impressions. It will be important for future work to experimentally manipulate partisan threat to establish it as a causal mechanism.

Salient behavioral information elicits updated face impressions [15]. The current findings show that disclosed partisanship is salient enough to elicit impression change, and this effect is pronounced among people with strong political ideologies and perceptions of partisan threat. Indeed, Experiment 1 showed partisan disclosure effects only among more liberal perceivers, but only more liberal perceivers reported perceiving Republicans as especially threatening. When more liberal *and* conservative perceivers evaluated opposing partisans as threatening in Experiment 2, partisan disclosure effects on face impressions emerged across perceivers. Experiment 2 also showed that the simply labeling people as sharing partisanship elicits more positive impressions almost immediately after evaluating faces, consistent with ingroup favoritism when membership is arbitrarily determined [37–39].

Although ideology effects on face impressions were paralleled by partisan threat perception effects across experiments, it is worth considering why inconsistencies across experiments might emerge. One previously discussed possibility regarded college students self-reporting being more conservative than they are when ideology is more objectively assessed [56]. Potential conflicts between self-reported and actual ideologies could lead to inconsistencies both within- and across-experiments. Speculatively, more objective ideology assessments could, in part, resolve inconsistencies. It could be that factors that are beyond the scope of the current work interfaced with perceived threat and ideology to relate to impressions. For example, people who have high actual [68] or even imagined [69] contact with opposing partisans have less affective polarization, findings that broadly reflect work on intergroup contact to reduce prejudice [70]. Future work may consider the extent to which relative partisan contact or isolation interfaces with perceived threat to affect face impressions separably or interactively.

The current work has broad implications for partisan interactions. Impressions of faces affect countless behaviors [e.g., 33]. Notably, outgroup disclosure quickly elicits avoidance tendencies [71] that are likely related to the communicative hesitation promoting intergroup tension [72]. The current work raises the possibility that this tension may reflect, in part, negative impressions of faces from people who perceive opposing partisans as especially threatening. Indeed, more negative impressions of faces are theorized to reflect a motivation to avoid them [67]. Speculatively, some people's more negative impressions of faces disclosed as opposing partisans may perpetuate overall intergroup tensions. Future work can disentangle the relationship between political ideology and partisan threat, by experimentally manipulating threat perceptions. This work would examine whether threat is a core feature of ideology or if there are contexts where ideological differences do not coincide with partisan threat and its pernicious consequences.

An open question regards whether polarized impressions based on political ideology can be changed. Indeed, to the extent that valence is a fundamental perception of face evaluation [73] relating to countless interpersonal outcomes [e.g., 19], more positive face impressions may be necessary to mitigate growing political sectarianism in the United States [16]. Future work may identify positive behavioral cues that are enough to counteract opposing partisan cues to begin addressing this possibility. For example, given that people place different weight on positive and negative morality- and competence-related behavioral information when updating impressions [74], one potential area for fruitful work would be to determine how behaviors in different domains may mitigate negative impressions of opposing partisans. Other work aimed creating more equitable partisan interactions may consider interventions that address impressions of faces [e.g., 75] and assess how longstanding intervention effects may be.

The current work also raises interesting avenues for future basic person perception research. Because people have stereotypic visualizations of group members [76], for example, disclosed partisanship may change impressions only to the extent to which faces match the stereotypic prototypes held by perceivers. Indeed, relative partisanship is often interpretated as

reflecting group divisions [7,25–27]. To further characterize how disclosed partisanship affects face impressions, future work can vary the characteristics of faces disclosed as partisans (e.g., trustworthy or untrustworthy) and address disclosure effects using both implicit and explicit measures. Such manipulations can clarify the strength of disclosure on impressions and at what levels they manifest. Moreover, it would be worthwhile to test how changing party affiliations or knowledge of a target's within-party disagreement affects face impressions. It could be that partisanship polarizes impressions only to the extent that partisans are perceived as being loyal to their party.

Simply labeling people as political partisans shifts impressions of their faces. These findings have implications for when people might disclose their partisanship to others. Based on Experiment 2, for example, people might avoid negative impressions by not disclosing their partisanship until they are in an inclusive space and perceived as relatively non-threatening. Affecting initial impressions of faces may be an initial step by which disclosing political partisanship affect countless aspects of social interactions, illustrating one way by which political partisanship shapes social cognition.

## Supporting information

**S1 File.**
(DOCX)

## Author Contributions

**Conceptualization:** Brittany S. Cassidy, Colleen Hughes, Anne C. Krendl.

**Data curation:** Brittany S. Cassidy, Colleen Hughes, Anne C. Krendl.

**Formal analysis:** Brittany S. Cassidy, Colleen Hughes.

**Funding acquisition:** Anne C. Krendl.

**Investigation:** Brittany S. Cassidy, Colleen Hughes.

**Methodology:** Brittany S. Cassidy, Colleen Hughes, Anne C. Krendl.

**Project administration:** Brittany S. Cassidy.

**Resources:** Anne C. Krendl.

**Supervision:** Anne C. Krendl.

**Validation:** Brittany S. Cassidy.

**Visualization:** Brittany S. Cassidy, Colleen Hughes.

**Writing – original draft:** Brittany S. Cassidy.

**Writing – review & editing:** Colleen Hughes, Anne C. Krendl.

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
