## [Decision Letter · Decision Letter 0]

1 Jul 2022

PONE-D-22-08989Disclosing Political Partisanship Polarizes First Impressions of FacesPLOS ONE

Dear Dr. Cassidy,

Thank you for submitting your manuscript to PLOS ONE. After careful consideration, we feel that it has merit but does not fully meet PLOS ONE’s publication criteria as it currently stands. Therefore, we invite you to submit a revised version of the manuscript that addresses the points raised during the review process.

We look forward to receiving your revised manuscript.

Kind regards,

Peter Karl Jonason

Academic Editor

PLOS ONE

Journal Requirements:

2. Please note that according to our submission guidelines (http://journals.plos.org/plosone/s/submission-guidelines), outmoded terms and potentially stigmatizing labels should be changed to more current, acceptable terminology. To this effect,  “Caucasian” should be changed to “white” or “of [Western] European descent” (as appropriate).

Reviewers' comments:

Reviewer's Responses to Questions

**Comments to the Author**

1. Is the manuscript technically sound, and do the data support the conclusions?

Reviewer #1: Partly

Reviewer #2: Yes

2. Has the statistical analysis been performed appropriately and rigorously? 

Reviewer #1: Yes

Reviewer #2: Yes

3. Have the authors made all data underlying the findings in their manuscript fully available?

Reviewer #1: Yes

Reviewer #2: Yes

4. Is the manuscript presented in an intelligible fashion and written in standard English?

Reviewer #1: Yes

Reviewer #2: Yes

5. Review Comments to the Author

Reviewer #1: The present work examines a critical question with important social implications: whether disclosing political affiliation impacts impressions drawn from faces. The authors present two studies that read methodologically stringent and strong. Overall, I enjoyed reading the paper and think that this work is worth getting published eventually. However, I refrain from making this recommendation at this point without suggesting some major revisions, mostly on the analyses and the interpretation of the results. Below you can find a list of reasons.

Major issues

Theoretical

-The difference between the past work by Mallinas et al. (2018) and the present work is not clear enough. Why would examining the same question in a romantic relationship context be a “limitation”? Is there a reason why the observed effect (e.g., negative attitudes towards those of opposing ideology) may or may not be generalized across different contexts? Perhaps the authors can highlight other procedural differences (consideration of the veracity of political ideology, changes in evaluations, etc.) across that past work and their work early on, as the current framing does not make these differences clear.

-The conceptualization of political identification gets obscure at times. Sometimes they interpret identification in terms of extremity (see page 7, though page numbers are missing on the initial pages). But they are measuring identification in terms of the strength of affiliation, which is clearly different from the extremity. I would suggest clarifying their definition of the concept and staying consistent throughout the paper.

Methods

- Some methodological details were missing, making it challenging for me to interpret the task in detail, especially in Experiment 1. The authors cited an osf page for additional methods, but I could not locate any files about methods on that page.

For instance:

-Which software was used to program the studies?

-How were picture pairs constructed in Experiment 1? Were the pairs randomly selected from lists of Republicans and Democrats for each participant, or were the pairs pre-determined and the same for all participants?

-How did they test recognition of each face (familiarity) after the task in Experiment 1?

-Exclusion criteria were vague for both experiments. How did they determine “not following the task instructions”?

Results

-I think the most major issues were about the chosen analyses and the interpretation of the results.

1. The description of the models made it a bit difficult to comprehend. As far as I understood, in Experiment 1, the DV was a choice (binary; republican = 0, liberal = 1), and the trait of evaluation (competency, likability; how that was coded is unclear) was a predictor. Is that correct? How were the predictors coded specifically? Were they dummy coded, effect coded, etc.? How was political ideology standardized? Was it mean-centered? Specifying all these either in the text or on the table would help with interpreting the results.

2. As the codings are not specified, I have trouble interpreting the tables. For instance, political ideology has a positive estimate of choice (DV). It is mentioned that democrat was coded as 1 and republican as 0 when choosing. I am confused because if higher ideology scores indicate more conservatism, were conservatives more likely to choose democrats as more competent/likable overall? As estimated marginal means suggest otherwise, there should be something I am missing here. The authors go straight to interpreting the post hoc tests without explaining the main tests here. I believe the main analyses require explanation.

3. The motivation behind adding the Trait as a predictor to the model is unclear. Did the authors have any predictions about trait differences? If competence and likability choices are strongly related (it seems so), it would be justifiable to average these scores and create a simpler model. Simplifying the model this way could also help with the convergence of random slopes (which were not included in the veracity analyses, perhaps that was because of a convergence issue?) and would also make a new (missing) model with the perceived threat as another predictor possible and more interpretable (see the analysis suggested below in point 6).

4. The veracity analysis in Experiment 1 is left uninterpreted. What does the significant main effect of veracity in Table 3 mean? Does it mean that the option labeled as “democrat “is evaluated as more competent overall (democrat labeled as democrat > democrat labeled as republican)?

5. The relevance of the partisan affiliation analyses to the study's main purpose was unclear. It is not surprising that ideology relates to relevant partisanship. I would move these analyses to supplementary materials and not interpret them in terms of face perception, as face perception is not part of these models.

6. Partisan threat could have been relevant to the study's main question. Yet again, the threat was analyzed separately from the face perception data, though the findings were interpreted in terms of its relevance to face perception. A direct test of the partisan threat’s role in face choice is missing in both experiments

7. Another standing question: were participants more likely to evaluate the faces of ingroup members (same ideology) as more likable/competent than those of outgroup members (other ideology) when ideology was not disclosed? The authors mention past work suggesting that in the introduction, but they do not report their findings on this question. Veracity was only analyzed among the disclosed ideology condition. Also, the title of these analyses reads a bit misleading: “Characterizing partisan disclosure effect on face impressions by veracity,”: but the manipulation of disclosure (nondisclosed vs. disclosed) was not a predictor in the reported analyses (only disclosed conditions are included).

Discussion

-The authors speculate about the relative roles of partisan affiliation and perceived threat in face impressions. Again, their data should allow them to analyze such relative effects. Why are these analyses unavailable? Perhaps, the studies are underpowered for such analyses, but if that is the case, the authors should at least comment on that. Then, I would recommend not interpreting the results in terms of their parallel to a perceived threat (as there are no direct analyses, these interpretations are too speculative) or, more ideally, conducting a third study to test the relationship between perceived threat and face perception directly.

Typos

P 7. “would the complement”

p. 21 “conservates”

Reviewer #2: In this article, the authors show that providing political identification information shifts initial evaluations (experiment 1) and updating (experiment 2) of explicit competence/likability ratings.

In my review, I have tried to adhere to the following guidelines of plos one “Unlike many journals which attempt to use the peer review process to determine whether or not an article reaches the level of 'importance' required by a given journal, PLOS ONE uses peer review to determine whether a paper is technically rigorous and meets the scientific and ethical standard for inclusion in the published scientific record.”

In my opinion, the present paper clearly meets the above standard of being technically rigorous and ethical. I go through each of the PLOS one criteria point-by-point, then conclude with some final thoughts

1. The study presents the results of primary scientific research.

2. Results reported have not been published elsewhere.

The present manuscript clearly meets the above 2 criteria

3. Experiments, statistics, and other analyses are performed to a high technical standard and are described in sufficient detail.

The present manuscript appears to follow all best practices of complex mixed effects models, and I was encouraged that they used random effects for both subject and stimuli (as suggested by the recent literature that they cite). I have no doubts of the technical integrity of their findings. If anything, they err on the side of reporting too much, and I think they could move some of the less important tables or statistics to a supplement (e.g., it’s probably not necessary to fully report how republicans and democrats vary on political ideology), and I think by carefully considering which analyses are central to their point they could streamline their paper.

4. Conclusions are presented in an appropriate fashion and are supported by the data.

They are. I would suggest three changes for Figure 1 to increase clarity: first, include what -1 and +1 standard deviation on political ideology refers to (so readers don’t have to scroll all the way back to the methods). Second, I think the graph may be clearer if they used facets only for likability vs. competence, and not for political ideology, which could become the x variable, with condition becoming the grouping variable (so, in other words, use aes(x = ideology, color = disclosure)). I think this would make all the values much closer together and easier to compare. Finally, consider including significance bars to highlight which cells are significantly different from one another.

5. The article is presented in an intelligible fashion and is written in standard English.

The article is quite intelligible.

6. The research meets all applicable standards for the ethics of experimentation and research integrity.

I have no reason to doubt the ethicality of the present research

7. The article adheres to appropriate reporting guidelines and community standards for data availability.

The article exceeds these, having both data and code already available. I was further encouraged that the posted code appears to be quite clean and commented well, which is uncommon (especially before a manuscript is accepted).

Together, I believe the present article meets the standards of PLOS One as I understand them. Some final suggestions to the authors for future research that may be of importance to the field:

First, if the authors have reaction times from the first study, it would be interesting to do a drift-diffusion model on the reaction times to see if the presence of ideology is shifting (a) the starting point bias, (b) the rate of accumulation, or both. This might further get at the mechanism of what is going on (i.e., are participants simply requiring less evidence to say the politically consistent individual is competent/likable, or are they accumulating evidence more steeply from individuals who share their ideology?).

Second, the updating question is interesting, and I would be interested to see it followed up with (a) implicit measures, particularly if they deviate from explicit measures, and (b) looking at more nuanced cases, such as finding someone switched political parties. It seems that learning about party affiliation, and how that shifts evaluations, is a potentially fruitful area of future research.

6. PLOS authors have the option to publish the peer review history of their article (what does this mean?). If published, this will include your full peer review and any attached files.

Reviewer #1: No

Reviewer #2: No

---

## [Author Response · Author response to Decision Letter 0]

16 Aug 2022

Responses to Individual Reviewer Concerns:

Reviewer 1

Comment 1: The difference between the past work by Mallinas et al. (2018) and the present work is not clear enough. Why would examining the same question in a romantic relationship context be a “limitation”? Is there a reason why the observed effect (e.g., negative attitudes towards those of opposing ideology) may or may not be generalized across different contexts? Perhaps the authors can highlight other procedural differences (consideration of the veracity of political ideology, changes in evaluations, etc.) across that past work and their work early on, as the current framing does not make these differences clear.

Action taken: Good point. We have revised the manuscript in several ways to highlight how it builds on past (related) work. The key difference between this manuscript and past work is that past work has not really considered why partisan cues elicit polarized impressions. In our revised manuscript, we leverage new included analyses (also suggested by the reviewer) to more concretely state that a secondary goal of our manuscript was to determine what effects co-occur with ideology effects on polarized impressions of faces that might help to better explain and characterize them. We first make this distinction on page 4 in the introduction, calling back to it on page 7 of the introduction and mentioning it further throughout the discussion sections of each experiment and the general discussion.

Comment 2: The conceptualization of political identification gets obscure at times. Sometimes they interpret identification in terms of extremity (see page 7, though page numbers are missing on the initial pages). But they are measuring identification in terms of the strength of affiliation, which is clearly different from the extremity. I would suggest clarifying their definition of the concept and staying consistent throughout the paper.

Action taken: We agree with the reviewer on this point. To this end, we have removed wording suggesting that we interpret political identification in terms of extremity from the manuscript. Instead, we use the simpler (and intended) definition of political identification as simply where partisans fall on a continuous ideological scale that we define on p. 10-11 in the Experiment 1 method.

Comment 3: Some methodological details were missing, making it challenging for me to interpret the task in detail, especially in Experiment 1. The authors cited an osf page for additional methods, but I could not locate any files about methods on that page.

Action taken: We now explicitly state the name of the file (Supplementary Information.docx) on OSF that provides additional methods information and results on p. 9.

Comment 4: Which software was used to program the studies? 

Action taken: We now state on p. 9 that we used E-Prime 2.0 to program the experiments.

Comment 5: How were picture pairs constructed in Experiment 1? Were the pairs randomly selected from lists of Republicans and Democrats for each participant, or were the pairs pre-determined and the same for all participants?

Action taken: We address these questions in the Experiment 1 method on p. 9. We write, “One hundred ten pairs of neutrally expressive Caucasian male faces were drawn from databases of opponents in United States political races that have been used in past work (e.g., Todorov et al., 2005). Each pair depicted one actual Republican and one actual Democrat who were opponents in a past political race. Thus, the pairs were pre-determined and the same across participants.”

Comment 6: How did they test recognition of each face (familiarity) after the task in Experiment 1?

Action taken: We clarify how we tested recognition on p. 10. We write, “Immediately after the task, participants disclosed if they recognized any faces. If they said yes (N = 65), disclosed who they thought they recognized. Our a priori exclusion criterion was to exclude any participants who accurately identified faces. No participants, however, did so.”

Comment 7: Exclusion criteria were vague for both experiments. How did they determine “not following the task instructions”?

Action taken: We clarify exclusion reasons for Experiment 1 on p. 9. We write, “Of 185 undergraduates recruited from a large Midwestern university in the United States, we excluded five. Two did not complete the partisanship characterization measures (see below), two failed the manipulation check (see below), and one did not report age.” We clarify exclusion reasons for Experiment 2 on p. 24. We write, “Of 101 undergraduates recruited from a large Midwestern university, we excluded seven for failing the manipulation check.”

Comment 8: The description of the models made it a bit difficult to comprehend. As far as I understood, in Experiment 1, the DV was a choice (binary; republican = 0, liberal = 1), and the trait of evaluation (competency, likability; how that was coded is unclear) was a predictor. Is that correct? How were the predictors coded specifically? Were they dummy coded, effect coded, etc.? How was political ideology standardized? Was it mean-centered? Specifying all these either in the text or on the table would help with interpreting the results.

Action taken: We clarify the model in Experiment 1 on p. 12 by writing, “We first tested whether disclosing partisanship more strongly affected impressions than non-disclosed partisanship. Likability and competency choices (Republican = 0, Democrat = 1) were logistically regressed on Trait (competent = 0, likable = 1), Task Version (disclosed labels = 0, non-disclosed labels = 1), Perceiver Political Ideology (standardized around the composite political ideology scores for the sample to have a mean of 0 and a standard deviation of 1), and their interactions as fixed effects.”

We clarify the model in Experiment 2 on p. 24 by writing, “Likability evaluations were regressed on Time (before label = 0, after label = 1), Partisan Label (Dummy coded using “undecided” as the reference of 0: Republican, Democrat, undecided), Perceiver Political Ideology (standardized as in Experiment 1), and their interactions as fixed effects.”

Comment 9: As the codings are not specified, I have trouble interpreting the tables. For instance, political ideology has a positive estimate of choice (DV). It is mentioned that democrat was coded as 1 and republican as 0 when choosing. I am confused because if higher ideology scores indicate more conservatism, were conservatives more likely to choose democrats as more competent/likable overall? As estimated marginal means suggest otherwise, there should be something I am missing here. The authors go straight to interpreting the post hoc tests without explaining the main tests here. I believe the main analyses require explanation.

Action taken: This comment refers to the Experiment 1 results section. Note that the political ideology effect in Experiment 1 is actually an odds ratio rather than the traditional slope recognized from linear regressions. The odds ratio of .85 actually means that as perceivers’ ideological conservatism increased, their likelihood of selecting Democrat faces decreased. 

As suggested by the reviewer, we now detail main effects underlying the higher order interaction of primary interest in Experiment 1. Instead of going straight to the post hoc tests that explain the interaction, we first write on p. 12, “Main effects of Task Version (reflecting more selected Democrats in the disclosed versus non-disclosed task version) and Perceiver Political Ideology (reflecting fewer selected Democrats with higher perceiver conservatism) were qualified by a Task Version × Perceiver Political Ideology interaction that partially supported this hypothesis (Table 1a; Fig.1). We allow main effects to be interpreted within the context of this higher-order interaction.”

Comment 10: The motivation behind adding the Trait as a predictor to the model is unclear. Did the authors have any predictions about trait differences? If competence and likability choices are strongly related (it seems so), it would be justifiable to average these scores and create a simpler model. Simplifying the model this way could also help with the convergence of random slopes (which were not included in the veracity analyses, perhaps that was because of a convergence issue?) and would also make a new (missing) model with the perceived threat as another predictor possible and more interpretable (see the analysis suggested below in point 6).

Action taken: Our motivation for adding Trait as a predictor to the model was largely conceptual. Warmth and competence reflect the “big two” dimensions of social perception, and comprise separable ways in which people stereotype others as detailed by the stereotype content model. Although we might not expect findings to differ across traits, their separable distinct relevance in the social perception literature makes their inclusion in the model likely of interest to a variety of researchers. We write on p. 8, “These traits were selected because they are core dimensions of person perception (Fiske et al., 2007) reflecting separable ways in which people stereotype others (Fiske et al., 2002). Devaluing these traits would complement distinct ways of deriding opposing partisans that are becoming more commonplace in the United States (Finkel et al., 2020).”

Comment 11: The veracity analysis in Experiment 1 is left uninterpreted. What does the significant main effect of veracity in Table 3 mean? Does it mean that the option labeled as “democrat “is evaluated as more competent overall (democrat labeled as democrat > democrat labeled as republican)?

Action taken: We now write on p. 16, “Main effects of Disclosed Label Veracity (reflecting fewer selected Democrats with accurate versus inaccurate labels) and Perceiver Political Ideology (reflected fewer selected Democrats with higher perceiver conservatism) emerged. Note that the Disclosed Label Veracity effect likely reflects the ideological skew of the sample. Suggesting disclosed labels affected impressions irrespective of their veracity, however, no interaction between Disclosed Label Veracity and Perceiver Political Ideology emerged.”

Comment 12: The relevance of the partisan affiliation analyses to the study's main purpose was unclear. It is not surprising that ideology relates to relevant partisanship. I would move these analyses to supplementary materials and not interpret them in terms of face perception, as face perception is not part of these models.

Action taken: Based on this comment and a comment from Reviewer 2, we have moved these analyses to the supplemental material document available on OSF.

Comment 13: Partisan threat could have been relevant to the study's main question. Yet again, the threat was analyzed separately from the face perception data, though the findings were interpreted in terms of its relevance to face perception. A direct test of the partisan threat’s role in face choice is missing in both experiments 

Action taken: This point is well taken. To address it, we include additional analyses for Experiments 1 (p. 20) and 2 (p. 30-31) that replace perceiver political ideology with the difference in perceived partisan threat from Democrats relative to Republicans in the primary models. Critically, using these threat scores produces virtually the same patterns of results across experiments. By including this direct test of the relation between perceived partisan threat and partisan disclosure effects on face impressions, we can more confidently suggest that perceiver ideology affects partisan disclosure effects on face impressions to the extent of threat perceived from different partisan groups.

Comment 14: Another standing question: were participants more likely to evaluate the faces of ingroup members (same ideology) as more likable/competent than those of outgroup members (other ideology) when ideology was not disclosed? The authors mention past work suggesting that in the introduction, but they do not report their findings on this question. Veracity was only analyzed among the disclosed ideology condition. Also, the title of these analyses reads a bit misleading: “Characterizing partisan disclosure effect on face impressions by veracity,”: but the manipulation of disclosure (nondisclosed vs. disclosed) was not a predictor in the reported analyses (only disclosed conditions are included).

Action taken: We address these questions in the results and discussion sections of Experiment 1. We write on p. 16 in the results section, “Because people are often able to detect partisanship from faces alone (e.g., Rule & Ambady, 2010), our next analyses concerned determining whether the veracity of disclosed labels affected polarized face impressions based on perceiver political ideology. First, we examined whether face impressions were polarized by non-disclosed partisanship. Among participants for whom party labels were not disclosed, however, perceiver political ideology did not affect face selections, OR = .98, p = .33, 95% CI [.94, 1.02].”

We write on p. 20-21, “Although partisanship can be detected from facial cues alone (e.g., Rule & Ambady, 2010), non-disclosed partisanship did not polarize face impressions. Although it could be that participants did not detect partisanship from these faces, another possibility is that being asked to evaluate traits overrode undisclosed partisanship effects on face impressions in this task overall (see Todorov et al., 2005). Future research may assess this possibility by addressing partisanship effects on face impressions when people are informed, for example, that they are evaluating politicians versus not.”

Comment 15: The authors speculate about the relative roles of partisan affiliation and perceived threat in face impressions. Again, their data should allow them to analyze such relative effects. Why are these analyses unavailable? Perhaps, the studies are underpowered for such analyses, but if that is the case, the authors should at least comment on that. Then, I would recommend not interpreting the results in terms of their parallel to a perceived threat (as there are no direct analyses, these interpretations are too speculative) or, more ideally, conducting a third study to test the relationship between perceived threat and face perception directly.

Action taken: We have removed language referring to the relative role of partisan affiliation from the manuscript. We now include new analyses (see response to Comment 13 of Reviewer 1) that allow us to interpret the results in terms of their parallel to a perceived threat. Finally, we refer to the potential experiment suggest by the reviewer in the General Discussion on p. 34 by writing, “Future work may disentangle the relationship between political ideology and partisan threat, perhaps by experimentally manipulating threat perceptions. This work would examine whether threat is a core feature of ideology or if there are contexts where ideological differences do not coincide with partisan threat and its pernicious consequences.”

Comment 16: Typos: P 7. “would the complement” 

p. 21 “conservates” 

Action taken: These typos have been corrected.

Reviewer 2

Comment 1: If anything, they err on the side of reporting too much, and I think they could move some of the less important tables or statistics to a supplement (e.g., it’s probably not necessary to fully report how republicans and democrats vary on political ideology), and I think by carefully considering which analyses are central to their point they could streamline their paper.

Action taken: Based on this comment and a similar comment from Reviewer 1, we have relegated analyses less central to our main point the supplemental material available on OSF to streamline the paper. For example, we moved analyses on how Republicans and Democrats vary on political ideology to supplemental information. We also moved analyses regarding affiliation ratings to the supplemental information.

Comment 2: I would suggest three changes for Figure 1 to increase clarity: first, include what -1 and +1 standard deviation on political ideology refers to (so readers don’t have to scroll all the way back to the methods). Second, I think the graph may be clearer if they used facets only for likability vs. competence, and not for political ideology, which could become the x variable, with condition becoming the grouping variable (so, in other words, use aes(x = ideology, color = disclosure)). I think this would make all the values much closer together and easier to compare. Finally, consider including significance bars to highlight which cells are significantly different from one another.

Action taken: We have edited Figure 1 to incorporate all changes suggested by the reviewer. We now include what -/+ 1 SD on political ideology refers to on the figure itself. We use facets only for likability and competence. Finally, we include significance bars to highly the cells that are significantly different from one another.

Comment 3: First, if the authors have reaction times from the first study, it would be interesting to do a drift-diffusion model on the reaction times to see if the presence of ideology is shifting (a) the starting point bias, (b) the rate of accumulation, or both. This might further get at the mechanism of what is going on (i.e., are participants simply requiring less evidence to say the politically consistent individual is competent/likable, or are they accumulating evidence more steeply from individuals who share their ideology?).

Action taken: This is a fantastic idea and one that we are considering as we continue this line of work. We bring up this idea in Experiment 1 Discussion on p. 21 by writing, “Future work may replicate these findings while focusing on reaction times to better understand why they emerged. For example, it could be that label disclosure enables people to require less evidence to say a similar relative to opposing partisans are competent and likable. However, it could also be that people more steeply accumulate evidence of competence and likability from similar relative to opposing partisans. Disentangling these findings using drift diffusion modeling (e.g., Johnson et al., 2017), can help clarify processes underlying face impressions polarized by partisanship.

Comment 4: Second, the updating question is interesting, and I would be interested to see it followed up with (a) implicit measures, particularly if they deviate from explicit measures, and (b) looking at more nuanced cases, such as finding someone switched political parties. It seems that learning about party affiliation, and how that shifts evaluations, is a potentially fruitful area of future research.

Action taken: Also great ideas! We address them in the General Discussion on p. 35 by writing, “To further characterize how disclosed partisanship affects face impressions, future work can vary the characteristics of faces disclosed as partisans (e.g., trustworthy or untrustworthy) and address disclosure effects using both implicit and explicit measures. Such manipulations can clarify the strength of disclosure on impressions and at what levels they manifest. Moreover, it would be worthwhile to test how changing party affiliations or knowledge of a target’s within-party disagreement affects face impressions. It could be that partisanship polarizes impressions only to the extent that partisans are perceived as being loyal to their party.”

---

## [Decision Letter · Decision Letter 1]

2 Sep 2022

PONE-D-22-08989R1Disclosing Political Partisanship Polarizes First Impressions of FacesPLOS ONE

Dear Dr. Cassidy,

Thank you for submitting your manuscript to PLOS ONE. After careful consideration, we feel that it has merit but does not fully meet PLOS ONE’s publication criteria as it currently stands. Therefore, we invite you to submit a revised version of the manuscript that addresses the points raised during the review process. Only minor issues remain. Please submit your revised manuscript by Oct 17 2022 11:59PM. If you will need more time than this to complete your revisions, please reply to this message or contact the journal office at plosone@plos.org. Please include the following items when submitting your revised manuscript:A rebuttal letter that responds to each point raised by the academic editor and reviewer(s). You should upload this letter as a separate file labeled 'Response to Reviewers'.A marked-up copy of your manuscript that highlights changes made to the original version. You should upload this as a separate file labeled 'Revised Manuscript with Track Changes'.An unmarked version of your revised paper without tracked changes. You should upload this as a separate file labeled 'Manuscript'.If applicable, we recommend that you deposit your laboratory protocols in protocols.io to enhance the reproducibility of your results. Protocols.io assigns your protocol its own identifier (DOI) so that it can be cited independently in the future. For instructions see: https://journals.plos.org/plosone/s/submission-guidelines#loc-laboratory-protocols. Additionally, PLOS ONE offers an option for publishing peer-reviewed Lab Protocol articles, which describe protocols hosted on protocols.io. Read more information on sharing protocols at https://plos.org/protocols?utm_medium=editorial-email&utm_source=authorletters&utm_campaign=protocols.

We look forward to receiving your revised manuscript.

Kind regards,

Peter Karl Jonason

Academic Editor

PLOS ONE

Journal Requirements:

Reviewers' comments:

Reviewer's Responses to Questions

**Comments to the Author**

1. If the authors have adequately addressed your comments raised in a previous round of review and you feel that this manuscript is now acceptable for publication, you may indicate that here to bypass the “Comments to the Author” section, enter your conflict of interest statement in the “Confidential to Editor” section, and submit your "Accept" recommendation.

Reviewer #1: All comments have been addressed

2. Is the manuscript technically sound, and do the data support the conclusions?

Reviewer #1: Yes

3. Has the statistical analysis been performed appropriately and rigorously? 

Reviewer #1: Yes

4. Have the authors made all data underlying the findings in their manuscript fully available?

Reviewer #1: Yes

5. Is the manuscript presented in an intelligible fashion and written in standard English?

Reviewer #1: Yes

6. Review Comments to the Author

Reviewer #1: As I mentioned in my previous review, I enjoyed reading this work and think that it offers some valuable contributions to the work on impression formation and updating.

I have now read the revision and the authors’ responses to both reviewers. I appreciate that the authors addressed the reviewers’ inquiries overall. I can now see their analytical approach in a much clearer way, and I believe the manuscript improved as a result. There are, however, a few standing issues for me before recommending the work for publication:

1. I appreciate that the reviewers added new analyses to examine the direct relationship between partisan disclosure effects and perceived partisan threat. The results suggest that, consistently across the two studies, the partisan threat effect is always in line with the perceiver ideology effects. That is, in Study 1, conservative ideology does not bring disclosure effects, nor does perceiving Democrats as more threatening (which correlates with conservative ideology). In Study 2, conservative ideology does bring disclosure effects, and so does perceiving Democrats as more threatening.

However, I am concerned that the authors interpret these results as if they are showing the partisan threat is the mechanism underlying the disclosure effects. For instance, when responding to Reviewer 1 Comment 13:

“By including this direct test of the relation between perceived partisan threat and partisan disclosure effects on face impressions, we can more confidently suggest that perceiver ideology affects partisan disclosure effects on face impressions to the extent of threat perceived from different partisan groups.”

Also, when they talk about their studies’ particular contribution on p. 4, they suggest that past research has not examined the mechanism, which implies their research does:

“[…] one limitation is that they do not explain why this polarization occurs.”

My concern about this interpretation is for two reasons. First, I would recommend avoiding making any claims about the mechanism without an experimental design or at least testing the mediation (ideally longitudinally but at least in a cross-sectional exploratory test). Second, it seems like some of their findings indeed contradict a mechanism interpretation. If the partisan threat is the mechanism underlying the disclosure effects on liking a particular face, shouldn’t partisan threat impact participants' liking of a particular (ingroup vs. outgroup) face similarly across liberal and conservative perceivers? That does not seem to be the case in Study 1. Perceiving Democrats as more threatening (mostly conservative participants, given the significant correlation) does not impact face perception, whereas perceiving Republicans as more threatening (mostly liberal participants, given the significant correlation) does. It seems like, even when a conservative perceiver feels threatened by a liberal (and they do to some degree, given the significant correlation), they do not report liking a liberal face less than a conservative face. As the threat effects are interpreted in comparison to different targets (perceiving a face from an ideological group as opposed to faces from other groups) rather than independently (the degree to which the perceiver ideology relates to the perception of threat from a certain target), it seems to me that these nuances in the findings do not get the attention they deserve. All in all, I think the authors should refrain from emphasizing threat as a mechanism in their framing.

2. Given the above interpretation, their findings suggest that threat may not fully explain conservative participants’ indifference to faces in Study 1. Then what does? I think the authors should elaborate more on the potential alternative explanations other than the perceived threat. For example, can conservative participants be less attentive to the choosing task used in Study 1 and pick who is more likable basically randomly? Relatedly, I think the revised manuscript would benefit from discussing the inconsistent findings across the two studies a bit more in-depth in the general discussion.

3. The authors described the findings in the general discussion: “the more conservative participants in Experiment 2 may have been more likely to outwardly derogate democrats” (p. 31). Yes, that is what their findings already suggest, but why can this be the case? Again, the general discussion seems to lack a discussion for potential reasons.

More minor issues:

-“Prior work supports disclosed partisanship as more generally polarizing face impression.”

(p. 4): The baseline for comparison is missing here (“more” than what?):

-I think partisan cues, and especially the labels used in this work, are not minimal cues as claimed on p. 6. They indeed seem very salient.

-“Although people accurately detect political partisanship from faces…” (p. 7): this sentence reads a bit too deterministic; they "can" detect, or they "tend to" detect?

-It seems odd to drop a participant’s data entirely for not reporting their age in Study 1. What is the rationale behind this decision? Were the results replicated when all participants were included in the analyses (at least this one participant who did not fail the attention and manipulation check)?

-How many task versions were there in total (including counterbalancing versions)? That would be nice to include in the main text (although one can probably figure it out by browsing the data files).

-This interpretation was not clear to me: “Main effects of Disclosed Label Veracity (reflecting fewer selected Democrats with accurate versus inaccurate labels) and Perceiver Political Ideology (reflected fewer selected Democrats

with higher perceiver conservatism) emerged. Note that the Disclosed Label Veracity effect likely reflects the ideological skew of the sample” (p. 16). Checking the distribution reported in the supplementary materials, the sample seems to be a little bit skewed towards Democrats. But wouldn’t we expect the opposite of these results were due to this skewness (i.e., more selected Democrats with accurate labels)? Perhaps the authors can clarify that part a bit more in the text.

7. PLOS authors have the option to publish the peer review history of their article (what does this mean?). If published, this will include your full peer review and any attached files.

Reviewer #1: No

---

## [Author Response · Author response to Decision Letter 1]

4 Oct 2022

September 20, 2022

Dear Dr. Jonason,

Please consider this revised manuscript (PONE-D-22-08989R1) titled, “Disclosed Political Partisanship Polarizes First Impressions of Faces” co-authored with Dr. Colleen Hughes and Dr. Anne Krendl, for consideration as a research article in PLOS ONE. This is an original manuscript, has not been published, and is not under consideration for publication elsewhere. All data and code are available on OSF. A link is provided in the manuscript. As detailed below, we have revised our manuscript based on the remaining comments of one reviewer. Substantive changes are denoted in the main text in red font. 

We are, of course, happy to make additional changes to the paper with the goal of improving the work. I thank you and the reviewers for your time and attention to this manuscript.

Sincerely,

Brittany Cassidy

Assistant Professor

Department of Psychology

University of North Carolina at Greensboro

bscassid@uncg.edu

Responses to Individual Reviewer Concerns:

Reviewer 1

Comment 1: The reviewer had a broad remaining concern that perceived threat should not be discussed as a mechanism for polarized face impressions based on disclosed partisanship because the manuscript provides no experimental evidence of it. The reviewer also notes that inconsistencies regarding perceived partisan threat both within- and across-experiments should be addressed in discussion sections and in the general discussion. The reviewer suggested a broader discussion of potential mechanisms for face impressions polarized by partisanship, especially in the general discussion.

Action taken: These were points well taken. We absolutely agree with the reviewer that causality must not be inferred when only correlational evidence for a relation can be presented. We have revised our manuscript in several ways to address this concern. 

First, we have removed language speaking to causality throughout the manuscript except when we suggest that future work use experimental manipulations of, for example, perceived partisan threat to establish a causal link with polarized face impressions (e.g., p. 4, 34). 

Second, we now discuss inconsistencies in perceived threat findings within- and across-experiments, being cautious to avoid redundancy. For example, we write in the Experiment 1 Discussion on p. 23, “Although threat ratings suggested that more conservative perceivers did not find Democrats as more threatening than Republicans, significant correlations emerged between perceiver ideology and partisan threat perceptions. What might have caused this inconsistency? One possibility may lie in the college-aged sample recruited for the experiment. College-aged students often show a bias to perceiving themselves as more conservative than they really are (Zell & Bernstein, 2014). If the students identifying themselves as more conservative were indeed more liberal than they realized, it would allow for the possibility of threat perceptions less extreme than those of the students identifying as more liberal. Indeed, these biased perceptions of one’s own partisanship are more pronounced for conservatives than for liberals (Zell & Bernstein, 2014).”

We also address these inconsistencies while considering new potential mechanisms in the General Discussion on p. 35 by writing, “Although ideology effects on face impressions were paralleled by partisan threat perception effects across experiments, it is worth considering why inconsistencies across experiments might emerge. One previously discussed possibility regarded college students self-reporting being more conservative than they are when ideology is more objectively assessed (Zell & Bernstein, 2014). Potential conflicts between self-reported and actual ideologies could lead to inconsistencies both within- and across-experiments. Speculatively, more objective ideology assessments could, in part, resolve inconsistencies. It could be that factors that are beyond the scope of the current work interfaced with perceived threat and ideology to relate to impressions. For example, people who have high actual (Whitt et al., 2021) or even imagined (Warner & Villamil, 2017) contact with opposing partisans have less affective polarization, findings that broadly reflect work on intergroup contact to reduce prejudice (Pettigrew, 1998). Future work may consider the extent to which relative partisan contact or isolation interfaces with perceived threat to affect face impressions separably or interactively.”

Comment 2: “Prior work supports disclosed partisanship as more generally polarizing face impression.”

(p. 4): The baseline for comparison is missing here (“more” than what?):

Action taken: We have rewritten the sentence for clarity. It now reads (p. 4), “Prior work supports that disclosed partisanship may polarize face impressions across contexts.”

Comment 3: I think partisan cues, and especially the labels used in this work, are not minimal cues as claimed on p. 6. They indeed seem very salient.

Action taken: We agree and have rewritten the sentence for clarity. It now reads (p. 6), “Such patterns would extend work showing favoritism and, sometimes, derogation, based on group membership (Brewer, 1999; Tajfel, 1970; Tajfel et al., 1971; Tajfel & Turner, 1979) from a romantic (Mallinas et al., 2018) to a more general context and show that simple partisan labels in the absence of other partisan information can powerfully affect impressions.

Comment 4: “Although people accurately detect political partisanship from faces…” (p. 7): this sentence reads a bit too deterministic; they "can" detect, or they "tend to" detect?

Action taken: We have rewritten the sentence to read (p. 8), “Although people can often detect political partisanship from faces (Rule & Ambady, 2010), disclosed group labels can override naturally occurring ones to elicit biases (Van Bavel et al., 2008).”

Comment 5: It seems odd to drop a participant’s data entirely for not reporting their age in Study 1. What is the rationale behind this decision? Were the results replicated when all participants were included in the analyses (at least this one participant who did not fail the attention and manipulation check)?

Action taken: We now report analyses from Experiment 1 that include the participant who did not report his/her age. None of the conclusions or statistical significance of the results changed.

Comment 6: How many task versions were there in total (including counterbalancing versions)? That would be nice to include in the main text (although one can probably figure it out by browsing the data files).

Action taken: We now clarify that there were three versions of Experiment 1 (p. 10) and three versions of Experiment 2 (p. 26).

Comment 7: This interpretation was not clear to me: “Main effects of Disclosed Label Veracity (reflecting fewer selected Democrats with accurate versus inaccurate labels) and Perceiver Political Ideology (reflected fewer selected Democrats with higher perceiver conservatism) emerged. Note that the Disclosed Label Veracity effect likely reflects the ideological skew of the sample” (p. 16). Checking the distribution reported in the supplementary materials, the sample seems to be a little bit skewed towards Democrats. But wouldn’t we expect the opposite of these results were due to this skewness (i.e., more selected Democrats with accurate labels)? Perhaps the authors can clarify that part a bit more in the text.

Action taken: We have clarified this interpretation in the Experiment 1 Results section on p. 16 by writing, “A main effect of Perceiver Political Ideology reflected fewer selected Democrats with higher perceiver conservatism. A main effect of Disclosed Label Veracity reflected fewer selected Democrats with accurate versus inaccurate labels. This pattern may seem surprising because given the liberal skew of the sample (see Supplemental Material), one might expect more accurately labeled Democrats because potential detections would match labels. Because differences between faces signal the likelihood of winning (Todorov et al., 2005), it could also be that inaccurate labels resulted in more “Democrats” with positively interpreted facial cues.”

---

## [Editor Report · Decision Letter 2]

6 Oct 2022

Disclosing Political Partisanship Polarizes First Impressions of Faces

PONE-D-22-08989R2

Dear Dr. Cassidy,

We’re pleased to inform you that your manuscript has been judged scientifically suitable for publication and will be formally accepted for publication once it meets all outstanding technical requirements.

Kind regards,

Peter Karl Jonason

Academic Editor

PLOS ONE
---

## [Editor Report · Acceptance letter]

14 Oct 2022

PONE-D-22-08989R2 

Disclosing Political Partisanship Polarizes First Impressions of Faces 

Dear Dr. Cassidy:

I'm pleased to inform you that your manuscript has been deemed suitable for publication in PLOS ONE. Congratulations! Your manuscript is now with our production department. 

Kind regards, 

on behalf of

Dr. Peter Karl Jonason 

Academic Editor

PLOS ONE